# Replacement of Fish Meal by Defatted Yellow Mealworm (*Tenebrio molitor*) Larvae in Diet Improves Growth Performance and Disease Resistance in Red Seabream (*Pargus major*)

**DOI:** 10.3390/ani9030100

**Published:** 2019-03-19

**Authors:** Atsushi Ido, Atsushi Hashizume, Takashi Ohta, Takayuki Takahashi, Chiemi Miura, Takeshi Miura

**Affiliations:** 1Graduate School of Agriculture, Ehime University, 3-5-7, Tarumi, Matsuyama, Ehime 790-8566, Japan; ido@agr.ehime-u.ac.jp (A.I.); hashiaezwebnejp@gmail.com (A.H.); takataka@dpc.ehime-u.ac.jp (T.T.); c.miura.6u@cc.it-hiroshima.ac.jp (C.M.); 2South Ehime Fisheries Research Center, Ehime University, 1289-1, Funakoshi, Ainan, Ehime 798-4292, Japan; ot1059@gmail.com; 3Department of Global Environment Studies, Faculty of Environmental Studies, Hiroshima Institute of Technology, 2-1-1 Miyake, Saeki, Hiroshima 731-5193, Japan

**Keywords:** yellow mealworm, fish meal replacement, red seabream, *Edwardsiella tarda*

## Abstract

**Simple Summary:**

Yellow mealworm is a potential novel protein source for sustainable food production, especially for aquaculture. In this study, the intake of a diet including defatted mealworm larvae was compared with a control diet containing fish meal from anchovy in a feeding trial with red seabream. As a result, the growth of red seabream fed the diet including defatted mealworm larvae with complete replacement of fish meal was higher than that of fish fed the control diet. Moreover, red seabreams fed diets including mealworm larvae gained disease resistance against pathogenic bacteria. The defatting process is thought to be important for insect-based diets, and the potential functional benefits for cultured fish from the diets, such as acquiring disease resistance, are remarkable.

**Abstract:**

Yellow mealworm (*Tenebrio molitor*) larvae are a potential alternative animal protein source for sustainable aquaculture. However, reports on the successful complete substitution of fish meal with yellow mealworm larvae in an aquaculture diet have been limited. In this study, we conducted a feeding trial with red seabream (*Pagrus major*) being fed diets with partial or complete replacement of fish meal with yellow mealworm larvae defatted with a hexane–ethanol solution. Feed intake in red seabream increased in accordance with yellow mealworm larvae inclusion, and diets including 65% defatted mealworm larvae (complete replacement of fish meal) showed significant growth promotion. The addition of the oil fraction from mealworm larvae to diets resulted in growth reduction, despite meeting the nutritional requirements of red seabream. Moreover, the survival rate of red seabreams fed diets with partial replacement of fish meal with mealworm larvae was significantly higher in a challenge test with pathogenic *Edwardsiella tarda* bacteria. The present study demonstrated that yellow mealworm larvae are not merely an alternative animal protein, but have potential as functional feed ingredients for aquaculture production.

## 1. Introduction

Due to the rapid expansion of global aquaculture production, the development of alternative protein sources in feed for carnivorous fishes has become quite an urgent issue. Insects have attracted broad attention as a novel protein source for enhancing global food security since the Food and Agriculture Organization of the United Nations (FAO) published an assessment of edible insects as food and feed [1]. 

Some studies have reported the potential as an alternative feed ingredient of yellow mealworm larvae (MW; *Tenebrio molitor* Latreille) (Tenebrionidae: Coleoptera) in aquaculture production [1,2,3,4]. Although this insect appears to possess adequate amino acid profiles for use in fish feed [1,2,3,5,6], reports on the successful complete substitution of fish meal (FM) with MW have been limited. Based on the evidence published to date, a diet with 50% to 70% replacement of FM with dried full-fat mealworm larvae resulted in similar growth to the FM diet in gilthead seabream (*Sparus aurata*) [7], European seabass (*Dicentrarchus labrax*) [8], black spot seabream (*Pagellus bogaraveo*) [9], and rainbow trout (*Oncorhynchus mykiss*) [10,11], and only 12% replacement for FM was achieved with defatted mealworm larvae in pearl gentian grouper (*Epinephelus lanceolatus* × *Epinephelus fuscoguttatus*) [12]. However, the MW’s potential in diets for red seabream (*Pagrus major* Temminck and Schlege) (Perciformes: Sparidae) has not been investigated.

Recently, immunostimulation activity via the dietary intake of insects in fish feed has drawn attention to the use of insects as feed ingredients [13]. A low inclusion of housefly (*Musca domestica*) pupae increased phagocytic activity and disease resistance against *Edwardsiella tarda* in red seabream (*Pargus major*) [14]. Dietary inclusion of MW is also known to increase the enzyme activities of the immune systems of European seabass [15], rainbow trout [16], mandarin fish (*Siniperca scherzeri*) [17], and pearl gentian grouper [12].

In this current study, we report that the complete replacement of fish meal with defatted MW promoted growth, and partial replacement enhanced disease resistance in the marine carnivorous fish, red seabream. Our study shows that MW is not only a protein source for aquaculture feed, but likely has functions for health optimization in cultured fish.

## 2. Materials and Methods 

### 2.1. Feed Ingredients

Two types of dried MW larvae meal, DMW-1 and DMW-2, were used in the growth trial and the challenge test, respectively. For the growth test, microwave-oven-dried yellow mealworm (MW) larvae reared with wheat bran and vegetable waste were obtained from Shintoa Corporation (Tokyo, Japan) for the growth test. To remove the oil fraction, the ground MW larvae were each suspended with a 4-fold volume of a solution containing 9 volumes of normal hexane and 1 volume of ethanol (99%) and then incubated overnight at room temperature with occasional gentle agitation. The supernatant was removed after centrifugation and the sediment was heated at 60 °C for 1 day or more until completely air-dried. This defatting process was repeated several times until the content of crude fat reached values below 8% on a dry basis; DMW-1 was thus obtained. The oil fraction from MW (MW oil, MO) was prepared from the aforementioned supernatant containing hexane/ethanol (9:1) by evaporation at room temperature until the volume was decreased by approximately 10-fold, and then the temperature was gradually increased from 40 to 80 °C until the residual organic solvents had evaporated. For the challenge test, DMW-2 produced in Shangdong, China was obtained from Shintoa Corporation. Fish meal from Peruvian anchoveta (*Engraulis ringens* Jenyns) comprising 65% crude protein and cod fish oil was also obtained from Shintoa Corporation (Tokyo, Japan). Docosahexaenoic acid (DHA) was kindly provided by Bizen Chemical Co., Ltd. (Akaiwa, Japan).

### 2.2. Experimental Diets

We produced two experimental diets: one for a growth test (Table 1) and the other for a challenge test (Table 2). For the growth test, five experimental diets were formulated. A control diet (65% FM) contained the following protein sources: 65% FM and 8% corn gluten meal. Three experimental diets contained graded levels of DMW-1 to replace 38%, 60%, or 100% of FM in the control diet, corresponding to dietary DMW-1 inclusion levels of 250 (25% MW), 400 (40% MW), and 650 g/kg (65% MW), respectively. Additionally, to assess the effect of the oil fraction from MW (MO), a diet containing 1.84% DHA to be equivalent in n-3 highly unsaturated fatty acid (HUFA) content to the 65% MW diet and 5.16% MO to replace the fish oil in the 65% MW diet was formulated (65% MW + MO). For the challenge test, three experimental diets were formulated. A control diet (50% FM) contained the following protein sources: 65% FM, 16% soybean meal, and 8% corn gluten meal. Two experimental diets contained graded levels of DMW-2 to replace 10% or 20% of the proteins from FM in the control diet, corresponding to dietary DMW-2 inclusion levels of 5% (5% MW) and 10% (10% MW), respectively. The crude protein level in these diets was targeted to be approximately 50% to obtain satisfactory growth of red seabream [18] in the growth test. Supplementation with plant-derived protein sources (soybean meal and corn gluten meal) was used to achieve this protein level. To prepare the diets for use in the feeding trials, powdered forms of all the components were thoroughly mixed and supplemented with fish oil, MW oil, or DHA. These mixtures were then granulated after adding water and air-dried at 60 °C for more than 1 day. The resulting diets were stored at 4 °C until use.

### 2.3. Feeding Trials

All animal experiments were carried out in accordance with the “Regulation for Animal Experiments at Ehime University.” The protocol was approved by the Institutional Animal Care and Use Committee (IACUC) of Ehime University (Permit Number: 3908). 

#### 2.3.1. Growth Test

Red seabreams were kindly provided by Yamasaki Giken co., Ltd. (Kochi, Japan) and were maintained in 1000 L tanks containing natural sea water in a flow-through system with sand filtration. Fish under one year of age with no traumatic injuries or malformations were included in the trials. Unique identification tags were intraperitoneally injected into the fish under anesthesia with 2-phenoxyethanol, and the animals were allocated into different feeding study groups of 32 fish (*n* = 32). Duplicate cultures per study group containing half of the fish were cultivated in 100 L tanks; thus, 10 tanks were used in the trial. In the trial, fishes were fed once per day except for Sunday with the diets listed in Table 1 to satiation. The water temperature was 17–20 °C in the trial. The trial was conducted over 4 weeks. The fork lengths and body weights (BW) of all fish were measured three times (at the beginning, after 2 weeks, and after 4 weeks) under anesthesia.

Fork length (FL) gain, body weight (BW) gain, FL gain rate, BW gain rate, specific growth rate, total feed intake per fish, and feed conversion ratio (FCR) were calculated as follows:FL gain (mm) = FL at trial end − initial FL,BW gain (g) = BW at trial end − initial BW,FL gain rate (%) = FL gain/initial FL × 100,BW gain rate (%) = BW gain/initial BW × 100,SGR, % day^−1^ = [(ln final BW – ln initial BW)/number of feeding days] × 100,Feed intake per fish (g) = total feed intake per group (g)/number of fish,FCR = total feed intake per group (g)/BW gain per group.

Measurements of individual fishes were used to obtain the FL, BW, FL gain, BW gain, FL gain rate, and BW gain rate in the feeding test groups, and values in the duplicate tanks were used to obtain the total feed intake per fish and feed conversion ratio in each study group.

#### 2.3.2. Challenge Test with *Edwardsiella tarda*

Another feeding trial in addition to that mentioned above was conducted for the challenge test with the fish pathogen *Edwasiella tarda*. Red seabreams provided by Yamasaki Giken co., Ltd. (Kochi, Japan) (mean weight 30. 4 g) which were never fed any MW were also used in the study. Twenty-two fish per tank and duplicate cultures per study group were cultured in 100 L tanks; thus, 6 tanks were used in the trial. Fish were cultivated for 56 days and were fed to satiation once per day, except for Sunday, with the diets listed in Table 2. *E. tarda* pathogens were obtained from the Fisheries Research Division, Ainan Town Office. *E. tarda* was isolated from a red seabream infected with *E. tarda* at an aquaculture site in Ainan, Ehime using Salmonella Shigella Agar. The bacterial cells were adjusted to 5 × 10^6^ cells/ml in Hank’s Balanced Salt Solution (HBSS). The fish in each group were challenged with an injection of 100 μl bacterial cell suspension into an intraperitoneal cavity. After the infection, deaths were recorded for 14 days and the survival rate was calculated. 

### 2.4. Proximate Composition, Amino Acid, and Fatty Acid Analysis

The proximate composition, amino acids, and fatty acids were analyzed using the Association of Analytical Communities (AOAC) methods [20]. The content of crude protein was analyzed using the Kjeldahl method. “Kjeltab” (containing K_2_SO_4_) was added to the ground samples, and the samples were digested in a block heater (Tecator TM Digestion Systems 2520, FOSS). The nitrogen content was automatically analyzed using an Auto analyzer (Kjeltec^TM^ 8400, FOSS). The nitrogen–protein conversion factor was 6.25 on the calculation of crude protein from the nitrogen content. Crude fat was analyzed using the Soxhlet extraction method. Petroleum ether was used as the solvent for extraction from the ground samples in an Automated extractor (Soxtec^TM^ 8000, FOSS). The content of ash was analyzed using an electric furnace (MMF-1, AS ONE).

Proteinogenic amino acids in the meal samples were analyzed using an automated amino acid analyzer (Shimadzu, Kyoto, Japan) after hydrochloric hydrolysis with sodium chloride. For methionine and cystine, the samples were oxidized with performic acid prior to hydrochloric hydrolysis. For tryptophan, samples were prepared with barium hydroxide octahydrate and thiodietylene glycol before hydrolysis with sodium chloride and analyzed with high-performance liquid chromatography. Taurine was measured using high-performance liquid chromatography. Fatty acids in samples were prepared with saponification and analyzed by gas chromatograph using a method described in food labelling standards by the Consumer Affairs Agency Japan (CAA, 2015). The analysis of proteiogenic amino acids, taurine, and fatty acids was conducted by the Japan Food Research Laboratories (Osaka, Japan).

### 2.5. Statistical Analysis

Statistically significant differences between the control and test groups were identified by Steel–Dwass multiple comparison tests as a post hoc test after a Kruskal–Wallis test for growth performance and by a log rank test with a Bonferroni correction for survival rate in the challenge test. Both tests were conducted with “R” software (https://www.r-project.org).

## 3. Results

### 3.1. Analysis of Amino Acids and Fatty Acids in Yellow Mealworm Larvae

We first analyzed the proximate compositions and amino acid profiles of the two MW meals (DMW-1 and DMW-2). The crude protein and crude lipid levels in DMW-1 and DMW-2 were similar to those of FM owing to the defatting process. MW contained all of the indispensable amino acids found in FM (Table 3). The amino acid balance of MW was comparable to that of FM, and the indispensable amino acid levels in the control and test diets (Table 1) were sufficient to meet the dietary requirement in red seabream [21]. However, the quantity of taurine was only 0.1 mg/g in MW compared to 1.5 mg/g in FM. Lack of dietary taurine results in green liver syndrome and growth retardation [22,23,24]; thus, taurine would need to be added as a supplement in test diets.

The fatty acids in the dried yellow mealworm larvae before defatting to DMW-1 were also analyzed. The oil fraction of the larvae was used for the diets as MO; thus, the fatty acid composition in MO corresponded to that of the larvae. Marked differences were found between the fatty acid compositions of fish oil and yellow mealworm (Table 4). The total ω-3 fatty acid level was 1.7% in yellow mealworm larvae in comparison with 27.3% in fish oil. In addition, the only ω-3 fatty acid present in yellow mealworm larvae was linolenic acid (18:3 n-3). Notably, yellow mealworm larvae entirely lacked n-3 highly unsaturated fatty acids (n-3 HUFAs), which are designated as ω-3 fatty acids with 20 or more carbons and are considered essential nutrients for marine fish.

### 3.2. Growth Performances of Red Seabream Fed Diets with Increasing Replacement of FM with DMW-1

To evaluate the dietary effects of replacement of FM with MW in the diets of red seabream, a feeding trial was conducted. Partial or complete replacement of FM with DMW-1 in the diets of red seabream led to significantly increased FL compared to in fish fed with the 65% FM diets (Figure 1; Table 5). The BW gain of fish increased depending on the level of DMW-1 content in the experimental diets (Figure 1; Table 5). The feed intake increased with the higher MW inclusion, but there were no clear differences between the feed conversion ratios for any of the dietary groups (Table 5). In the 65% MW + MO group, the FL gain did not show any difference, but the BW gain rates, specific growth rates (SGR), and feed intake levels were significantly decreased compared with in the 65% MW group.

### 3.3. Challenge Test with Edwardsiella Tarda

We conducted a challenge test with *Edwardsiella tarda*, which has a high pathogenicity on red seabream. Fish were fed with the control diet or diets containing 5% or 10% DMW-2 for 56 days before the challenge. These diets did not lead to a significant effect on fish growth. After being challenged with *E. tarda*, fish fed the diet including DMW-2 showed a higher survival ratio than did the control, and diets containing 10% DMW-2 resulted in the highest survival rate (Figure 2).

## 4. Discussion

The present study shows that MW larvae are not only good sources of nutrients, but can also improve growth performance and anti-disease capacity in red seabream. The growth of fish fed with diets containing defatted MW (DMW-1) was significantly higher than that of those with control diets containing FM, and higher inclusion levels of DMW-1 in the diet (up to 65%, complete replacement of FM) resulted in larger growth in fish. On the other hand, anti-bacterial disease capacity of red seabream was obtained with only 10% inclusion of DMW-2 in the diet.

Although replacement of FM with MW has been attempted in many earlier studies, high inclusion levels of insects in aquaculture feed have resulted in growth reductions in many cases, particularly in marine carnivorous fish species [4]. For example, the growth performance with 50% full-fat MW in diets which was similar to that with FM diets was inferior to that with 25% MW inclusion in gilthead seabream [7], and 50% inclusion of full-fat MW negatively affected the growth performance comparing with FM diets in European seabass [8]. For black spot seabream, although 25% and 50% full-fat MW in diets did not significantly affect growth [9], the diet containing 50% MW also contained 31% FM; thus, complete replacement of FM was not achieved in their study. In rainbow trout, inclusion of 25%–50% full-fat MW (33% to 67% replacement of FM) resulted in an improvement in growth performance [10]. 

In contrast to the findings of the studies that utilized a partial replacement of FM, the present study showed that complete replacement of FM with MW in the diet of red seabream can have a positive impact on growth performance when the oil fraction is removed using organic solvents from dried MW. Intact MW larvae have excess crude lipids (30%–35%, dry matter); therefore, a defatting process was necessary for an adequate feed composition [6]. Chitin, a linear homopolymer of β(1–4)-linked N-acetylglucosamine units and a major constituent of insect cuticles, was thought to result in growth retardation in marine fish because of the difficulty in digesting it [26]. Although 6.6% of total nitrogen in MW larvae is estimated to be chitin nitrogen [27], chitin in diets could be digested and utilized in red seabream. Our findings are consistent with a previous study which found that the growth of red seabream was improved when diets were supplemented with 10% chitin [28]. 

Body weight gain increased in fish with defatted MW-inclusion diets, and the FCR in all groups was not significantly different. This suggests that defatted MW is remarkably preferred by red seabream. Notably, our current trials indicate that MW without the oil fraction promotes growth and feed intake in red seabreams compared a diet of MW with the oil fraction (MW+MO). Considering that the dietary n-3 HUFA level meets their nutrition requirements in both the 65% MW and 65% MW + MO diets, the oil fraction of MW was speculated to have a negative effect on fish feed intake from the result of the growth test in our study. Further studies are needed to evaluate the mechanism of the dietary intake reduction with the oil fraction of MW. Our results suggest that the defatting process in MW does not only decrease the fat level but also eliminates the negative effect of intact MW larvae inclusion in diets on fish feed intake. Therefore, eliminating the oil fraction is thought to be an important first step in the development of insect-based feeds for marine aquaculture.

Our previous study demonstrated that housefly (*Musca domestica*) pupae in diets increased immunostimulation activity and *E. tarda* resistance in red seabream [14]. Moreover, novel soluble polysaccharides with immunostimulation activity were identified from melon fly (*Bactrocera cucurbitae*) [29], Japanese oak silkmoth (*Antheraea yamamai*) [30], and silkworm (*Bombyx mori*) [31]. In another study, diets with low inclusion of black soldier fly (*Hermetia illucens*) larvae were reported to enhance immune activity and disease resistance against *Salmonella gallinarum* in broiler chicks [32]. 

In addition to the results with other insect species mentioned above, the dietary intake of MW is known to have immunostimulating activity in fish [15,16,17]. Moreover, a 7.5% MW inclusion was reported to enhance *Vibrio harveyi* resistance in juvenile pearl gentian grouper [12]. Although they concluded that chitin in MW could enhance the immune response in fish, the mechanism is still unknown. In the current study, red seabreams fed with diets including only 10% DMW-2 were thought to acquire *E. tarda* resistance via immunostimulation. Dietary intake of chitin or unknown polysaccharides in MW, such as Dipterose [27] or Silkrose [28,29], might play a role in anti-bacterial disease capacity in red seabream. *E. tarda* infection in red seabream aquaculture has been a serious problem; thus, diets including MW would not only provide an alternative dietary protein source, but would also increase disease resistance in fish.

## 5. Conclusions

We successfully conducted a feeding trial with red seabream (*Pagrus major*) being fed diets with complete replacement of fish meal with MW defatted with a hexane–ethanol solution, and the survival rate of red seabreams fed diets with partial replacement of fish meal with mealworm larvae was significantly higher in a challenge test with pathogenic *Edwardsiella tarda* bacteria. 

The present study demonstrated that yellow mealworm larvae are not merely an alternative animal protein, but have potential as functional feed ingredients for aquaculture production. Although MW is easy to rear with agricultural byproducts, optimization of the mass-rearing system is necessary to decrease the environmental impact for wider implementation [33]. MW production is less efficient than that of other insects, such as the black soldier fly or Argentine cockroach (*Blaptica dubia*) [34]. Diets including MW, however, could have functional benefits that no other insect species have. Further investigations on the benefits of dietary intake of MW, especially long-term feeding trials, are still needed to elucidate a method to reduce aquaculture cost.

## Figures and Tables

**Figure 1 animals-09-00100-f001:**
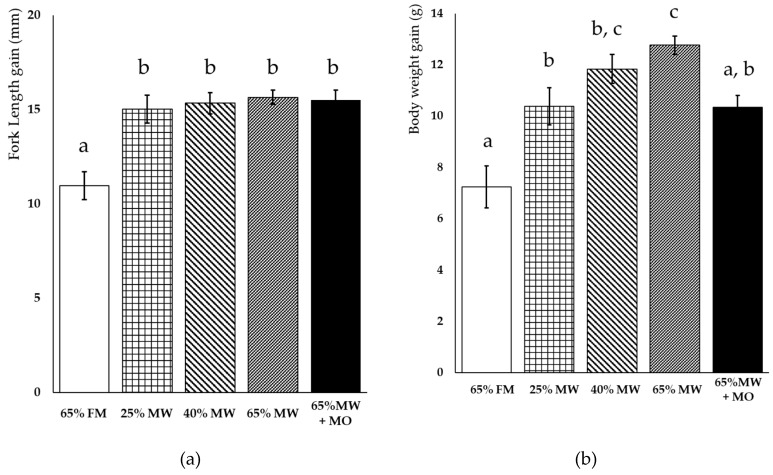
Growth performance of red seabream fed diets with a partial or complete replacement of FM with DMW-1 for a period of 4 weeks. Fork length gain (**a**) and body weight gain (**b**) at 4 weeks in the feeding trial. Values are means with their standard error represented by vertical bars [*n* = 29 in 65% FM, 31 in 25% MW, 32 in 40% MW, 32 in 65% MW, and 30 in 65% MW and the oil fraction of mealworm larvae (65% MW + MO)]. Different letters (a, b, c) indicate statistically significant differences according to the Steel–Dwass multiple comparison tests as a post hoc test after a Kruskal–Wallis test (*p* < 0.05).

**Figure 2 animals-09-00100-f002:**
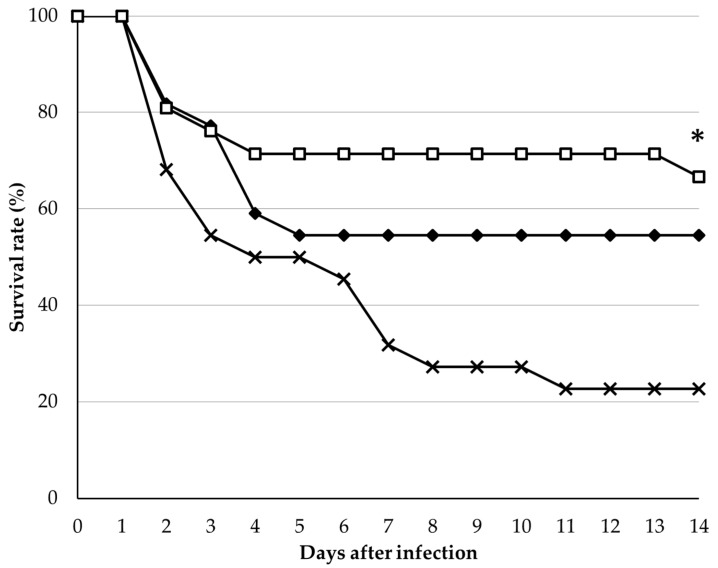
Survival rates of red seabream after infection with *Edwardsiella tarda*. Control diet group (X mark), 5% MW diet group (filled diamond), and 10% MW diet group (open square). The asterisk * indicates statistically significant differences compared with the control group using a log rank test with a Bonferroni correction (*p* < 0.05).

**Table 1 animals-09-00100-t001:** Formulation, proximate composition, and amino acid composition of the diets fed to red seabream for the growth test for 4 weeks.

**Ingredients (%)**	**65% FM ^a^**	**25% MW ^b^**	**40% MW ^c^**	**65% MW ^d^**	**65% MW+ MO ^e^**
Fish meal	65.00	40.00	25.00		
DMW-1		25.00	40.00	65.00	65.00
Fish oil	4.00	5.00	6.00	7.00	
Mealworm oil					5.16
DHA					1.84
Starch	12.00	14.00	15.00	17.00	17.00
Corn gluten meal	8.00	5.00	3.00		
Taurine	1.50	1.50	1.50	1.50	1.50
Vitamin mix	0.80	0.80	0.80	0.80	0.80
Mineral mix	0.40	0.40	0.40	0.40	0.40
Choline chloride	0.10	0.10	0.10	0.10	0.10
Vitamin C derivatives	0.10	0.10	0.10	0.10	0.10
NaH_2_PO_4_	0.80	0.80	0.80	0.80	0.80
KH_2_PO_4_	0.80	0.80	0.80	0.80	0.80
Calcium lactate	1.50	1.50	1.50	1.50	1.50
Carboxymethyl cellulose	5.00	5.00	5.00	5.00	5.00
Total	100.00	100.00	100.00	100.00	100.00
**Proximate composition (% on a dry matter basis) ^f^**					
Crude Protein	54.7	54.1	53.0	52.2	52.4
Crude fat	11.4	11.0	9.4	10.0	8.4
Ash	13.6	10.6	8.5	5.8	6.2
**Indispensable amino acids (% total amino acids) ^g^**					
Arg	6.2	6.0	5.9	5.7	5.7
His	3.2	3.2	3.2	3.2	3.2
Ile	4.3	4.4	4.5	4.6	4.6
Leu	9.0	8.6	8.2	7.8	7.8
Lys	7.5	6.9	6.5	5.9	5.9
Met + Cys	4.6	3.7	3.2	2.3	2.3
Phe	4.5	4.3	4.1	3.9	3.9
Thr	4.4	4.3	4.3	4.2	4.2
Trp + Tyr	2.2	3.1	3.6	4.4	4.4
Val	5.1	5.7	6.0	6.6	6.6

Abbreviations: DMW-1, defatted mealworm larvae with hexane and ethanol solvent; DHA, Docosahexaenoic acid.^a^ Control diet including 65% fish meal (FM).^b^ Test diet including 25% DMW-1 (38% replacement of FM in the control diet).^c^ Test diet including 40% DMW-1 (62% replacement of FM in the control diet).^d^ Test diet including 65% DMW-1 (100% replacement of FM in the control diet). ^e^ Test diet including 65% DMW-1 (100% replacement of FM in the control diet), and the oil fraction of mealworm larvae (MO) was added. DHA was supplemented to be equivalent in n-3 highly unsaturated fatty acid (HUFA) content to the 65% MW diet.^f^ Proximate composition on dry matter basis. Values are reported as means of duplicate analysis.^g^ The amounts of amino acids in the diets were calculated using the amino acid profiles in FM, DMW-1 (Table 3), and corn gluten meal [19].

**Table 2 animals-09-00100-t002:** Formulation of the diets fed to red seabream for the challenge test for 8 weeks.

Ingredients (%)	Control ^a^	5% MW ^b^	10% MW ^c^
Fish meal	50.00	45.00	40.00
DMW-2	0.00	5.00	10.00
Fish oil	6.00	6.25	6.50
Starch	6.00	6.00	6.00
Wheat meal	11.20	10.85	10.50
Soybean meal	16.00	16.00	16.00
Corn gluten meal	8.00	8.00	8.00
Taurine	0.00	0.10	0.20
Vitamin mix	0.40	0.40	0.40
Mineral mix	0.30	0.30	0.30
Choline chloride	0.05	0.05	0.05
Vitamin C derivatives	0.05	0.05	0.05
KH_2_PO_4_	1.00	1.00	1.00
Carboxymethyl cellulose	1.00	1.00	1.00
Total	100.00	100.00	100.00

Abbreviations: DMW-2, defatted mealworm larvae produced in Shangdong, China, obtained from Shintoa Corporation.^a^ Control diet including 50% fish meal.^b^ Test diet including 5% DMW-1 (10% replacement of FM in the control diet). ^c^ Test diet including 10% DMW-1 (20% replacement of FM in the control diet).

**Table 3 animals-09-00100-t003:** Proximate compositions and amino acid compositions of defatted mealworm larvae (DMW-1 and DMW-2) and fish meal for the experimental diets.

Proximate Compositions and Amino Acid Compositions	DMW-1	DMW-2	FM
Proximate composition (% on a dry matter basis) ^a^			
Crude protein	75.3	76.5	71.3
Crude fat	5.6	5.3	9.8
Ash	5.1	11.7	16.4
Amino acids (% total amino acids)			
Ala	7.4	5.8	7.0
Arg	5.7	7.0	6.5
Asp	8.7	8.3	9.5
Cys	1.0	3.2	1.0
Glu	12.8	13.4	13.2
Gly	5.6	6.8	7.7
His	3.2	2.1	3.3
Ile	4.6	3.8	4.3
Leu	7.8	8.5	8.2
Lys	5.9	7.0	8.3
Met	1.3	1.6	3.1
Phe	3.9	4.4	4.3
Pro	7.4	6.2	5.0
Ser	4.9	7.0	4.3
Thr	4.2	4.5	4.6
Trp	1.4	0.8	1.3
Tyr	7.6	3.5	3.3
Val	6.6	6.1	5.2
Taurine (mg/g dry matter)	0.1	0.1	1.5

Abbreviations: DMW-1, defatted mealworm larvae with hexane and ethanol solvent; DMW-2, defatted mealworm larvae produced in Shangdong, China, obtained from Shintoa Corporation. FM, fish meal obtained from Shintoa Corporation. Values of amino acids in FM were based on the FM in the Standard Table of Feed Composition in Japan [25]. ^a^ Proximate composition on a dry matter basis. Values are reported as means of duplicate analysis.

**Table 4 animals-09-00100-t004:** Fatty acid compositions in mealworm larvae and fish oil for the experimental diets.

Components (% Fatty Acids)	Dried Mealworm Larvae ^a^	Fish Oil
Saturated fatty acid		
12:0	0.3	4.1
14:0	2.8	-
15:0	0.2	0.5
16:0	15.5	15.6
17:0	0.2	0.7
18:0	2.4	3.6
20:0	-	0.3
22:0	-	-
Total	21.4	24.8
Monounsaturated fatty acid		
14:1	-	-
16:1	1.9	5.0
17:1	0.1	0.5
18:1	35.7	19.4
20:1	-	5.2
22:1	-	5.0
24:1	-	0.6
Total	37.7	35.7
Polyunsaturated fatty acid		
ω-3 fatty acid		
18:3n−3	1.7	1.0
20:3n−3	-	0.2
20:4n−3	-	0.7
20:5n−3	-	7.5
21:5n−3	-	0.3
22:5n−3	-	1.9
22:6n−3	-	15.7
Total	1.7	27.3
ω-6 fatty acid		
18:2n−6	35.7	2.8
20:2n−6	-	0.3
20:3n−6	-	0.2
20:4n−6	-	1.1
22:5n−6	-	0.6
Total	35.7	5.0
Others		
16:2	-	0.3
16:3	-	0.2
16:4	-	0.3
Total	-	0.8
Not identified	1.6	4.3

Lower limit of quantification. ^a^ Dried mealworm larvae before defatting to DMW-1 obtained from Shintoa Corporation.Lower limit of quantitation: 1 mg/g.

**Table 5 animals-09-00100-t005:** Initial fork length and body weight, fork length and body weight gain, SGR, feed intake, FCR, and survival rate of red seabream fed diets with a partial or complete replacement of FM with DMW-1 for a period of 4 weeks.

Parameters	65% FM	25% MW	40% MW	65% MW	65% MW+ MO
*n*	29	31	32	32	30
Initial FL (cm)	10.3 (10.1, 10.5) ^a^	10.3 (10.1, 10.4) ^a^	10.4 (10.2, 10.6) ^a^	10.4 (10.2, 10.6) ^a^	10.2 (10.0, 10.4) ^a^
FL gain (cm)	1.1 (0.9, 1.2) ^a^	1.5 (1.4, 1.7) ^b^	1.5 (1.4, 1.6) ^b^	1.6 (1.5, 1.6) ^b^	1.6 (1.4, 1.7) ^b^
Initial BW (g)	24.3 (22.8, 25.8) ^a^	24.4 (23.2, 25.6) ^a^	25.9 (24.5, 27.3) ^a^	25.4 (24.0, 26.7) ^a^	24.5 (23.0, 26.0) ^a^
BW gain (g)	7.2 (5.6, 8.9) ^a^	10.4 (8.9, 11.9) ^b^	11.8 (10.7, 13.0) ^b, c^	12.8 (12.0, 13.5) ^c^	10.4 (9.5, 11.2) ^a, b^
FL gain rate (%)	10.7 (9.2, 12.2) ^a^	14.7 (13.2, 16.2) ^b^	14.8 (13.6, 15.9) ^b^	15.1 (14.3, 15.9) ^b^	15.2 (14.1, 16.4) ^b^
BW gain rate (%)	26.9 (20.9, 32.8) ^a^	37.9 (32.7, 43.2) ^b, c^	39.2 (35.8, 42.5) ^b, c^	42.3 (40.2, 44.3) ^c^	36.6 (32.7, 40.6) ^a, b^
SGR (% day^-1^)	0.89 (0.69, 1.10) ^a^	1.27 (1.08, 1.46) ^b, c^	1.34 (1.23, 1.46) ^b, c^	1.46 (1.39, 1.54) ^c^	1.27 (1.16, 1.36) ^a, b^
Feed intake (g)	8.5	11.2	13.8	14.6	12.2
FCR	1.17	1.08	1.17	1.14	1.18
Survival rate (%)	90.6	96.9	100.0	100.0	93.8

Abbreviations: FL, fork length; BW, body weight; SGR, specific growth rate; FCR, feed conversion ratio. Data are represented by means (upper limit, lower limit of 95% confidential interval). Different letters indicate statistically significant differences according to the Steel–Dwass multiple comparison test as a post hoc test after a Kruskal–Wallis test (*p* < 0.05).

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
