# Peer review of "Replacement of Fish Meal by Defatted Yellow Mealworm (Tenebrio molitor) Larvae in Diet Improves Growth Performance and Disease Resistance in Red Seabream (Pargus major)"

_animals, 2019, doi:10.3390/ani9030100_

Round 1

Reviewer 1 Report

The revisions largely improved the manuscript. I have a minor suggestion:

Line 159-175: Please cite the scientific literature for the proximate composition and the amino acid analysis methods the authors use.

Author Response

Response to reviewer 1

The revisions largely improved the manuscript. I have a minor suggestion:

Line 159-175:

Please cite the scientific literature for the proximate composition and the amino acid analysis methods the authors use.

We wish to express our deep appreciation for your suggestion.

We add the citation of AOAC method “Official methods of analysis of AOAC International;” in L170-171.

Reviewer 2 Report

The changes and corrections requested have been partially done by the Authors in the new version, however many parts of the manuscript need a major revisions:

Abstract:  

Lines 27-28: Could you change the sentence “ In this study, we  conducted a feeding trial with red seabream (Pagrus major) for diets in complete replacement of fish  meal with yellow mealworm larvae defatted with a hexane-ethanol solution.” to “In this study, we conducted a feeding trial with red seabream (Pagrus major) fed diets with partial or complete replacement of fish meal with yellow mealworm larvae defatted with a hexane-ethanol solution.”

Introduction:

Lines 46,51,52,54,55, 58…. and through all the manuscript (including the references), could you put the scientific name of the animals in italic, ex. Tenebrio molitor.

Line 47: Could you replace “for use as fish feed” by for use in fish feed.

Line 48: …… “substitution of fish meal”, could you add the abbreviation (FM) and removed from line 49 in ….replacement for fish meal (FM).

Line 56: Could you remove the sentence “not only for substitution of FM”.

Line 64: Could you remove “(Pagrus major)”

Materials and methods:

Line 68 : 2.1. Feed Ingredients

Could you add this sentence, in order to make a clear statement that you used 2 type of MW for two different tests-> Two type of dried MW larvae meal, DMW-1 and DMW-2, were used in the growth trial and the challenge test, respectively. For the growth test:  a microwave oven-dried yellow mealworm larvae, reared on subtract containing wheat bran and vegetable waste, were obtained from Shintoa Corporation (Tokyo, Japan).

Lines 74-75: “This defatting process was repeated several times until the content of crude fat reached below 8% and the defatted MW (DMW-1) was obtained.” could you replace by “This defatting process was repeated several times until the content of crude fat reached values below 8% (on dry basis or wet weight, could specify ? ) and the DMW-1 was obtained.

Line 79:  Could you replace “Defatted MW  (DMW-2) produced in Shangdong, China was obtained from Shintoa Corporation for the challenge test.” with “For the challenge test, DMW-2 was obtained from Shintoa Corporation, produced in Shangdong, China”

Line 84: 2.2. Experimental diets

Who produced the experimental diets?

Line 86 : For the growth test, five experimental diets were formulated. A control diet (65% FM) contained protein sources 65% FM (and ??-> could you specify the rest of the protein source used in the experimental diet?).  Three experimental diets were formulated, contained graded levels of DMW-1 to replace 38, 60 and 100% of the proteins from FM in the control diet, respectively, and corresponding to dietary DMW-1inclusion levels of 250, 400 and 650 g/kg, respectively.  

Line 88 : “Additionally, to assess the effect of the oil fraction from MW (MO)” here you have to describe that is is the same as the diet 65% DMW-1 and the fish oil

The percentage given in the manuscript correspond to the inclusion level of 25, 40 and 100% in the diets and the partial and the complete replacement of FM by DMW-1 is about 38, 60 and 100%. Could you please check all these data and make the corrections.

Also the name that you give to your diets, are all different sometimes it is DMW-1 and in the table are MM, could you be consistent ?

Lines 90-92: the same comments as above, could you correct the percentage and be consistent with the name of the diets. saying that three experimental diets were formulated, where the control (describe the protein source with % (FM and vegetable proteins) and that the 2 diets you replaced 10 and 20% of the protein from FM with DMW-2.

Table 1

For the proximate composition, you have CP, CL and Ash, how about the content of dry matter?

For the essential amino acids composition, the units is % of crude protein or g/kg diet, could you precise please and also have the same name for DMW-1.

The abbreviations are confusing, I couldn’t see the letter a-e in the table, only f for the % of proximate composition.

Table 2, also same as above, could you check the % and the name given to the diets and also the abbreviations.

Feeding trials: how many tanks were used for the growth trial/condition?

Results  

Lines 184-190: Analyzed proximate composition and amino acids content of the feed ingredients, DMW-1, DMW-2 and FM, used in the diets of red seabream were …………could you describe the results if the content is similar and the differences between the three feed ingredients(re-write again the paragraph).

Table 3, give the % of dry matter

also give just the abbreviation in the table DMW-1… FM

why you didn’t give the content of the non-indispensable AAs of the 5 experimental diets used for the growth test.  

also re-write the title of  Table3

Table 4 Fatty acid composition of DMW (which one) and fish oil used in the experimental diets.

What is the limit of quantification of the fatty acids, could you specify?

Line 207: this is in the Table 3 legend? Thus, fatty acid composition in MO was the same as that of the larvae.

Line 214-215: could you reformulate this sentence “In an addition for the FL, BW gain in fish increased as more DMW-1 was contained in the diets (Figure 1; Table 5).”

Figure 1 legend: Growth performance of red seabream fed diets with a partial or complete replacement of FM with DMW-1 for a period of 4 weeks. Values are means with their standard deviation represented by vertical bars (n=?).

Table 5: Could you be consistent with the presentation of the results for feed intake, FCR and survival rate (upper limit and lower limit and also the statistics) and also describe the abbreviations for FL….

Also could you remove the columns of the week as you state before that the feeding trial was done during 4 weeks.

For the growth trial, could you give the values for the initial weight, does the fish had the same weight in the beginning of the trial and the calculated the specific growth rate? Also do you have values for the hepatosomatic and viscerosomatic index?

Line 232: do you have the results for the growth performances for the challenge test ?

Discussion:

It is very hard to understand the discussion as the English is very poor,

Example: The present study showed that the potential of MW for substitution of FM and for anti-disease function.  I couldn’t understand the meaning of this sentence, thus I suggest to the authors to send the manuscript to English language editing and improve the discussion.

Author Response

Reviewer 2

The changes and corrections requested have been partially done by the Authors in the new version, however many parts of the manuscript need a major revisions:

We wish to appreciate for your valuable comments. We improved the paper in accordance with your instructions, and the improved paper was edited by MDPI English Editing service. We highlighted the changes in underline in the revised manuscript.

Abstract:

Lines 27-28:

Could you change the sentence “ In this study, we conducted a feeding trial with red seabream (Pagrus major) for diets in complete replacement of fish meal with yellow mealworm larvae defatted with a hexaneethanol solution.” to “In this study, we conducted a feeding trial with red seabream (Pagrus major) fed diets with partial or complete replacement of fish meal with yellow mealworm larvae defatted with a hexaneethanol solution.”

We changed the sentence in accordance with your instruction.

Introduction:

Lines 46,51,52,54,55, 58…. and through all the manuscript (including the references), could you put the scientific name of the animals in italic, ex. Tenebrio molitor.

We carefully check the manuscript and confirm all scientific names in italic.

Line 47: Could you replace “for use as fish feed” by for use in fish feed.

We changed the word in accordance with your instruction.

Line 48: …… “substitution of fish meal”, could you add the abbreviation (FM) and removed from line 49 in ….replacement for fish meal (FM).

We changed the word in accordance with your instruction.

Line 56: Could you remove the sentence “not only for substitution of FM”.

We removed it as your instruction.

Line 64: Could you remove “(Pagrus major)”

We removed it as your instruction.

Materials and methods:

Line 68 : 2.1. Feed Ingredients

Could you add this sentence, in order to make a clear statement that you used 2 type of MW for two different tests> Two type of dried MW larvae meal, DMW1 and DMW2, were used in the growth trial and the challenge test, respectively. For the growth test: a microwave oven-dried yellow mealworm larvae, reared on subtract containing wheat bran and vegetable waste, were obtained from Shintoa Corporation (Tokyo, Japan).

As your instruction, we added “Two types of dried MW larvae meal, DMW-1 and DMW-2, were used in the growth trial and the challenge test, respectively. For the growth test,” in L70-71.

Lines 74-75:

“This defatting process was repeated several times until the content of crude fat reached below 8% and the defatted MW (DMW1) was obtained.” could you replace by “This defatting process was repeated several times until the content of crude fat reached values below 8% (on dry basis or wet weight, could specify ? ) and the DMW1 was obtained.

We changed the sentence into “This defatting process was repeated several times until the content of crude fat reached values below 8% on a dry basis, and DMW-1 was thus obtained”in L77-78.

Line 79: Could you replace “Defatted MW (DMW2) produced in Shangdong, China was obtained from Shintoa Corporation for the challenge test.” with “For the challenge test, DMW2was obtained from Shintoa Corporation, produced in Shangdong, China”

We changed the sentence in accordance with your instruction.

Line 84: 2.2. Experimental diets

Who produced the experimental diets?

We produce them, therefore we changed the sentence “We produced two experimental diets”in L88.

Line 86: For the growth test, five experimental diets were formulated. A control diet (65% FM) contained protein sources 65% FM (and ??>Could you specify the rest of the protein source used in the experimental diet?). Three experimental diets were formulated, contained graded levels of DMW1 to replace 38, 60 and 100% of the proteins from FM in the control diet, respectively, and corresponding to dietary DMW1inclusion levels of 250, 400 and 650 g/kg, respectively.

We changed the sentence into “A control diet (65% FM) contained the protein sources 65% FM and 8% corn gluten meal. Three experimental diets contained graded levels of DMW-1 to replace 38%, 60%, or 100% of the proteins from FM in the control diet, corresponding to dietary DMW-1 inclusion levels of 250 (25% MW), 400 (40% MW), and 650 g/kg (65% MW), respectively.” In L89-93.

Line 88 : “Additionally, to assess the effect of the oil fraction from MW (MO)” here you have to describe that is is the same as the diet 65% DMW1 and the fish oil The percentage given in the manuscript correspond to the inclusion level of 25, 40 and 100% in the diets and the partial and the complete replacement of FM by DMW1 is about 38, 60 and 100%. 

Could you please check all these data and make the corrections. Also the name that you give to your diets, are all different sometimes it is DMW1 and in the table are MM, could you be consistent?

We add the information as “a diet containing 1.84% DHA to be equivalent in n-3 highly unsaturated fatty acid (HUFA) content to the 65% MW diet and 5.16% MO to replace the fish oil in the 65% MW diet was formulated (65% MW + MO).”

We also check the data you pointed, and the names of the diets.

Lines 90-92:

the same comments as above, could you correct the percentage and be consistent with the name of the diets. saying that three experimental diets were formulated, where the control (describe the protein source with % (FM and vegetable proteins) and that the 2 diets you replaced 10 and 20% of the protein from FM with DMW2.

As your instruction, we added the sentence “three experimental diets were formulated. A control diet (50% FM) contained the protein sources 65% FM, 16% soybean meal, and 8% corn gluten meal. Two experimental diets contained graded levels of DMW-2 to replace 10% or 20% of the proteins from FM in the control diet, corresponding to dietary DMW-2 inclusion levels of 5% (5% MW) and 10% (10% MW), respectively.” in L96-99.

Table 1

For the proximate composition, you have CP, CL and Ash, how about the content of dry matter?

We specified “% on a dry matter basis” in table 1.

For the essential amino acids composition, the units is % of crude protein or g/kg diet, could you precise please and also have the same name for DMW1.

We change the unit of amino acids in table 1 into “% total amino acids” in accordance with that of table 3.

The abbreviations are confusing, I couldn’t see the letter ae in the table, only f for the % of proximate composition.

We put the letter, a-e in table 1.

Table 2, also same as above, could you check the % and the name given to the diets and also the abbreviations.

We checked them as your instruction, and put the letter, a-c.

Feeding trials: how many tanks were used for the growth trial/condition?

We specified the number of tanks we used in L140, and L160.

Results

Lines 184-190:

Analyzed proximate composition and amino acids content of the feed ingredients, DMW1, DMW2 and FM, used in the diets of red seabream were …………could you describe the results if the content is similar and the differences between the three feed ingredients (rewrite again the paragraph)

We changed the sentence into “We first analyzed the proximate compositions and amino acid profiles of the twoMW meals (DMW-1 and DMW-2). The crude protein and crude lipid levels in DMW-1 and DMW-2 were similar to those of FM owing to the defatting process.”in L195-197.

Table 3, give the % of dry matter also give just the abbreviation in the table DMW1… FM why you didn’t give the content of the nonindispensable AAs of the experimental diets used for the growth test. also rewrite the title of Table3

We removed the section “indispensable AA” and “dispensable AA” in table 3, because AA requirements were different among animals. In table 1, we showed indispensable AA as Forster (1998) showed. The dispensable AA in diets was not shown in table1, to avoid complexity. In other studies, only indispensable AA was shown, e.g. Song (2018), Sankian (2018), Piccolo (2017).

Table 4 Fatty acid composition of DMW (which one) and fish oil used in the experimental diets. What is the limit of quantification of the fatty acids, could you specify?

We specified the limit of quantification (1mg/g) in the footnotes of Table 4. As we wrote in L203, dried mealworm before defatting were analyzed.

Line 207: this is in the Table 3 legend? Thus, fatty acid composition in MO was the same as that of the larvae.

We moved the sentence to the Result section in L204-205.

Line 214-215: could you reformulate this sentence “In an addition for the FL, BW gain in fish increased as more DMW1 was contained in the diets (Figure 1; Table 5).”

We changed the sentence into “The BW gain of fish increased depending on the level of DMW-1 content in the experimental diets” in 229-230 

Figure 1 legend: Growth performance of red seabream fed diets with a partial or complete replacement of FM with DMW1 for a period of 4 weeks.

We corrected it as your instructions

Values are means with their standard deviation represented by vertical bars (n=?).

We added “Values are means with their standard error represented by vertical bars (n = 29 in 65% FM, 31 in 25% MW, 32 in 40% MW, 32 in 65% MW, and 30 in 65% MW+MO).” in L234-235

Table 5: Could you be consistent with the presentation of the results for feed intake, FCR and survival rate (upper limit and lower limit and also the statistics) and also describe the abbreviations for FL….

We checked them, and they are consistent with our result.

Also could you remove the columns of the week as you state before that the feeding trial was done during 4 weeks.

We removed it as your instruction.

For the growth trial, could you give the values for the initial weight, does the fish had the same weight in the beginning of the trial and the calculated the specific growth rate? 

We put “for period of 4 weeks” in the title of table 5. We added the initial weight and the specific growth rate. 

Also do you have values for the hepatosomatic and viscerosomatic index?

We agree that hepatosomatic and viscerosomatic index as you suggested would be valuable. Regrettably, however, because we did not have the data, we are unable to show them in table 5.

Line 232: do you have the results for the growth performances for the challenge test ?

The growth of the challenge test was indicated in L245. Regrettably, we did not get detailed data, because the purpose of the study is not for growth.

Discussion:

It is very hard to understand the discussion as the English is very poor, Example: The present study showed that the potential of MW for substitution of FM and for antidisease function. I couldn’t understand the meaning of this sentence, thus I suggest to the authors to send the manuscript to English language editing and improve the discussion.

We totally rewrite the Discussion section. The whole manuscript was edited with MDPI English editing service, thus the English were improved.

Round 2

Reviewer 2 Report

- The authors have responded to all the reviewers comments and I recommend this for publication after a few minor fixes:

Line 107: Could you add in the title of Table 1, the period of the growth test

Line 122: Could you add here also in the title of Table 2,  the period of the challenge test

Line 222: lower limit of quantification, could you add LLQ in all the empty space in the Table 4, where the fatty acids were not detected.          

Line 272: the growth performance with 50% full-fat MW in diets was inferior to that with 25% MW inclusion. Is it in gilthead seabream (7) or European seabass (8), could you please specify?

Line 277: In contrast to the findings of the studies that utilized replacement of FM-> it is a partial replacement? Could you added a partial replacement?

Line 279: …..when the oil fraction is removed using organic solvents-> could you add from MW-1

Line 288: …..the FCR between the  groups was not significantly different-> could you precise which groups ?

Line 293: was speculated to have a negative effect on fish feed intake.-> Do you have references showing that the oil fraction of MW could have a negative effect on feed intake ?

This manuscript is a resubmission of an earlier submission. The following is a list of the peer review reports and author responses from that submission.

Round 1

Reviewer 1 Report

The study by Ido et al, is evaluating the replacement of fish meal by defatted yellow mealworm in the diets of red seabream. The results show that this feed ingredient is a promising fish meal alternative for aquaculture feeds, while it seems to positively affect fish health and disease resistance.

The article overall was easy to follow, and the English is adequate, although the manuscript would benefit from a professional English editor to improve some grammar and expression.

Several points though require some attention:

-In the simple summary, a better description is required of the results, so please make this paragraph more accurate.

- In the introduction, if there is relevant information concerning defatted vs non-defatted meals please add it to justify your decision to do this step.

- In the materials and methods, please add references for all the methods the authors followed, I am largely missing this part.

Table 1: Replace ‘ether extract’ with ‘crude extract’

Line 105: Why did the authors perform their trials only in duplicates? The common practice is to have at least triplicate units of each treatment; usually, the individual variation is quite high.

Line 106: Is it a common practice to feed juvenile red seabream only once per day?

Line 121-128: Why did the authors test these dietary levels for the challenge trial? Please make it clear if you are referring in two different trials; the fish are the same in the two trials or the authors used fish that were never fed before with the MW? It is not quite clear. 

Please add a section referring in detail to the statistic methodology for the analysis of the results, I am missing this part, although relevant information exists in each figure.

-In the discussion, the authors should develop a bit more their discussion concerning their results, as I feel it is quite insufficient. Why do the authors think that the MO is inhibiting growth and feed consumption; do the authors hint what might be causing this immunostimulation effect?

Author Response

Response to reviewer 1

Thank you very much for admitting the significance of MS. We revised the MS referring to your comments. Please see following how we handled your comments. We highlighted the changes in underline in the revised manuscript.

The general comments:

The study by Ido et al, is evaluating the replacement of fish meal by defatted yellow mealworm in the diets of red seabream. The results show that this feed ingredient is a promising fish meal alternative for aquaculture feeds, while it seems to positively affect fish health and disease resistance. The article overall was easy to follow, and the English is adequate, although the manuscript would benefit from a professional English editor to improve some grammar and expression.

Response to the general comments:

Thank you for your careful review and valuable comments. 

Reviewer’s comment:-In the simple summary, a better description is required of the results, so please make this paragraph more accurate.

Our response:

     According to the reviewer’s comments, we additionally described the results in L19-20; “As a result, the growth of red seabream fed the diet including defatted mealworm larvae with complete replacement of fish meal was higher than that of fish fed the control diet.”

Reviewer’s comment:- In the introduction, if there is relevant information concerning defatted vs nondefatted meals please add it to justify your decision to do this step.

Our response:

    According to the reviewer’s comments, we precisely rewrote about the previous studies about full-fat (non-defatted) mealworm larvae [7-11], and about defatted meal worm larva [12] in L49-55. We emphasized that the potential of mealworm in diet s for red seabream (Pargus major) has not been disclosed. The necessity of defatting process was referred in L263-264; Intact MW larvae have an excess of crude lipids (30-35%, dry matter), therefore, a defatting process was necessary for an adequate feed composition [6].

Reviewer’s comment:- In the materials and methods, please add references for all the methods the authors followed, I am largely missing this part.

Our response:

    We rewrote the materials and methods section. Detailed compositions of the experimental diets were shown in 2.2. Experimental diets (L85-L91), details of Proximate composition, amino acid and fatty acid analysis were showed in 2.4.(L158-175), and 2.5. Statistical Analysis (L176-180) was added.

Reviewer’s comment: Table 1: Replace ‘ether extract’ with ‘crude extract’

Our response:

    We used “crude fat” in the replace of “ether extract”.

Reviewer’s comment: Line 105: Why did the authors perform their trials only in duplicates? The common practice is to have at least triplicate units of each treatment; usually, the individual variation is quite high.

Our response:

    Since we measured FL and BW of all fish with the unique identification tags (n=32) (L129), it enabled that FL and BW of each individual fish during the feeding trials was identified. Owing to the unique identification tags, we got FL gain and BW gain of each 32 fish for statistical analysis. Moreover, we conducted duplication cultures per experimental groups to confirm a reproducibility (L131). As we showed standard errors in FL and BW in Figure 1, the individual variation was not so high.

Reviewer’s comment: Line 106: Is it a common practice to feed juvenile red seabream only once per day?

Our response:

   Yes. We sufficiently fed juvenile red seabream by satiation (L132). Since juvenile red seabreams grow rapidly with feeding once a day, they are suitable for the growth test with experimental diets.

Reviewer’s comment: Line 121-128: Why did the authors test these dietary levels for the challenge trial? 

Our response:

  We conducted 2 feeding trials, one for the growth and the other for the challenge. Whereas the purpose of the feeding trials for growth was for complete replacement of fishmeal, a purpose t for the challenge was to find functional use of MW. In this purpose, we tried how few dietary intakes of MW gives anti-disease function.

Reviewer’s comment: Please make it clear if you are referring in two different trials; the fish are the same in the two trials or the authors used fish that were never fed before with the MW? It is not quite clear.

Our response:

   We clarified it in L148-149; “Another feeding trial other than the above mentioned was conducted for the challenge test with a fish pathogen Edwasiella tarda”.

Reviewer’s comment: Please add a section referring in detail to the statistic methodology for the analysis of the results, I am missing this part, although relevant information exists in each figure.

Our response:

   According to the reviewer’s comments, we added a section “2.5. Statistical analysis” in L176-180. “2.5. Statistical analysis - Statistically significant differences between the control and test groups were identified by Steel-Dwass multiple comparison tests as a post hoc test after a Kruskal-Wallis test for growth performance, and a log-rank test with a Bonferroni correction for survival rate in the challenge test. Both tests were conducted with “R” software (https://www.r-project.org).”

Reviewer’s comment: -In the discussion, the authors should develop a bit more their discussion concerning their results, as I feel it is quite insufficient. Why do the authors think that the MO is inhibiting growth and feed consumption; do the authors hint what might be causing this immunostimulation effect?

Our response:

  We carefully rewrote the discussion. We discussed the effects of chitin in mealworm larvae for the growth with a reference that growth of red seabream was improved with 10% chitin supplementation (Kono et al. “Effect of Chitin, Chitosan, and Cellulose as Diet Supplements on the Growth of Cultured Fish.” 1987) (L264-269). Moreover, we mentioned the necessity of further studies about the intake reduction with MO (L276-277), and the role of chitin or other polysaccharides for immunostimulation effect (L288-292). 

Reviewer 2 Report

The authors studied the effect of a partial or complete replacement of fish meal with defatted yellow mealworms larvae meal on growth performance and disease resistance in red seabream. The use of processed insect meal in farmed fish species is very interesting and important subject. Insect meal represent today a valuable alternative to the conventional proteins used in aquafeed. In addition, using dietary insect meals might improve fish health. Thus, this study is interesting, however many parts need to be implemented and clarified. In addition, the discussion somehow it gives me an impression that the authors were in hurry to complete the discussion section. 

The whole manuscript need a major revision. Please see bellow some comments:

Introduction:

Lines 43-44: “Some studies have reported the viability of yellow mealworm larvae (MW; Tenebrio molitor  Latreille) (Tenebrionidae: Coleoptera) in aquaculture production” -> Could you clarify this sentence, what do you mean by viability?

Line 47: “partial replacement of dried mealworm larvae resulted in similar growth as the FM” -> partial replacement of dried mealworm with what?”

Lines 51-52: “However, complete replacement of fish meal resulted in growth reduction of marine carnivorous fish in these studies.”-> in those cited studies, not all of them did a complete replacement of fish meal with mealworm meal, could you check these data ?

Lines 52-53: “Recently, the immunostimulation activity via dietary intake of insects in fish feed has drawn attention to the use of insects as feed ingredients”-> could you give some references, examples, is it the intake of insect that have an immunostimulation activity?

Lines 61-62: “but likely has functions for health management in cultured fish.”-> What does mean health management?

Material and methods:

It will be nice to start with

2.1. Feed ingredients

2.2. Experimental diets

2.3. Feeding trials

2.3.1 Growth performances

2.3.2 Challenge test with ..

2.4. Amino acid and fatty acid analysis

Of course the part about the statistics is missing in the material and methods.

Lines 71-73: “This defatting process was repeated several times until the crude lipids reached a comparable or lower level than fish meal and the defatted MW (MW meal) was obtained.”-> How did you compared the obtained defatted MW with the fish meal? Did you analyzed the feed ingredients, fish meal used and the most important the defatted mealworm?

What was the subtract used for the growth of MW?

Did you used another defatted MW for the challenge test, if yes could you give another name?

Lines 81-86: could you develop more about the amino acid and fatty acid analysis, the chemicals, the internal standard….

Line 88: could you stat and clarify that 2 experimental diets were produced, one for the growth of red seabreams (Table 1) and one for the challenge test (Table 2).

Line 87: In the part of diet composition, a lot of data are missing, like the inclusion level and presentation of the diets; 65% FM is the control group and after that the fish meal was partially or completely replaced by defatted mealworms at ….

The title of the tables are not clear and there are not a footnotes?

Example Table 1; Title: formulation, proximate composition of the diets fed to …..

Is it % on wet weight, dry ?

footnote: explain what is the 65% FM, 25% MW….. and DHA is used as abbreviation but not defined before in the text .

Why the non-essential amino acid taurine was added?

How about the methionine and lysine, as they are deficient in IM, did you add those essential amino acids in the experimental diets?

Line 100, could you remove artificially seed

the growth tests are missing a lot of information, example during which time the trial was done, how long,  the sampling how many, for each experience write again n=?

Again there are no information about the statistics analysis?

Results:

as you insect meal used in this study is defatted, maybe you could give a name like DMW.

Line 135: you analyzed the amino acid and the fatty acid composition of the defatted mealworms used as feed ingredients for the growth trial or for the challenge test?  You analyzed the AA and FA but not the protein and the fat content?

why did you use the reference number 6 ?

the fish meal in Table 3 is the same used to formulate the diets ?

The same with Table 4, the fatty composition of MW is the one used in the growth trial ? in addition the data of fish oil is it the one used as well in the experimental trial? how about the mealworm oil, did you analyze it also ?

Same comment for Table 3 and 4, the titles are not clear and the footnotes are missing.

Line 149: Growth performances of red seabream fed diets with increasing replacement of FM with DMW.

Line 151-155: “We tested 65% FM in comparison to 25% MW, 40% MW and 65% MW  diets (containing 38%, 62% and 100% replacement of FM, respectively). Additionally, to assess the effect of the oil fraction from MW (MO), 65% MW + MO diets were tested. DHA was supplemented in the 65% MW + MO to be equivalent in n-3 HUFAs content with the 65% MW diet.” -> it should be in the material and methods part.

Line 155-157:  “Diets containing MW showed a statistically significant tendency to promote FL and BW gain, depending on the increase in the MW content, with the 65% MW group showing the maximum growth (Figure 1; Table 156 5).” this sentence is unclear, if you could rephrase it, example a partial or complete replacement of FM with DMW in the diets of red seabream led to significantly increase the FL and BW compared to fish fed with the 65% FM diets. In addition for the BW, could explain also the differences observed between fish fed the % of DMW.

The title for the Figure 1 need to changed, example Growth performances of red seabream fed diets with increasing replacement of FM with DMW….

The same comment as before for the title of Table 5.

Table 5 and 6 could be in the same table. In Table 5 there are almost the same data as in Figure 1, I feel that there are repetition and no need to have the FL and BW three different times (with different calculation).

Line 178-180: if you could rephrase the sentence.

Figure2:  could you put the legend for the figure (x mark, filled diamond and Open Square) on the side of the figure.

Discussion:

My main concern is the discussion, as there is almost no discussion and this part need to be completed.

Line 190-191: “High rates of insects in aquaculture feed have resulted in growth reduction in many cases, particularly in carnivorous fish species.” -> this is not true, as now many papers showed that is possible to include high level of insect ingredients in the diets of carnivorous fish, such as salmonids, salmon and trout.

Line 194-195: could you provide references.

Line 201: might work negatively?????

Line 202: which negative effect from intact MW larvae?

The references need to be checked, italic for the Latin name

Author Response

Response to reviewer 2

Thank you very much for admitting the significance of MS. We revised the MS referring to your comments. Please see following how we handled your comments. We highlighted the changes in underline in the revised manuscript.

The general comments:

The authors studied the effect of a partial or complete replacement of fish meal with defatted yellow mealworms larvae meal on growth performance and disease resistance in red seabream. The use of processed insect meal in farmed fish species is very interesting and important subject. Insect meal represent today a valuable alternative to the conventional proteins used in aquafeed. In addition, using dietary insect meals might improve fish health. Thus, this study is interesting, however many parts need to be implemented and clarified. In addition, the discussion somehow it gives me an impression that the authors were in hurry to complete the discussion section.

Response to the general comments:

Thank you for your careful review and valuable comments. We have completely revised the manuscript in accordance with the reviewer’s comments. 

Reviewer’s comment:

Introduction:

Lines 43-44: “Some studies have reported the viability of yellow mealworm larvae (MW; Tenebrio molitor Latreille) (Tenebrionidae: Coleoptera) in aquaculture production” -> Could you clarify this sentence, what do you mean by viability?

 Our response:

     As your comment, we rewrote the sentence as “Some studies have reported the potential for alternative feed ingredients of yellow mealworm larvae (MW; Tenebrio molitor Latreille) (Tenebrionidae: Coleoptera) in aquaculture production”

Reviewer’s comment:

Line 47: “partial replacement of dried mealworm larvae resulted in similar growth as the FM” -> partial replacement of dried mealworm with what?”

Our response:

     We clarified what the dried mealworm larvae was replaced of, as “50 % to 70 % replacement for fish meal (FM) of dried full-fat mealworm larvae in diet resulted in similar growth as the FM diet.”

Reviewer’s comment:

Lines 51-52: “However, complete replacement of fish meal resulted in growth reduction of marine carnivorous fish in these studies.”-> in those cited studies, not all of them did a complete replacement of fish meal with mealworm meal, could you check these data ?

Our response:

   We checked the results of the studies again, and no study reported a complete replacement of fish meal with mealworm. Therefore, we precisely described these studies in L 49-55 and in L 251-258

Reviewer’s comment:

Lines 52-53: “Recently, the immunostimulation activity via dietary intake of insects in fish feed has drawn attention to the use of insects as feed ingredients”-> could you give some references, examples, is it the intake of insect that has an immunostimulation activity?

Our response:

We added a reference, Gasco, et al. “A. Can diets containing insects promote animal health?”J. Insects as Food Feed2018, a review showing studies about insects to optimize animal health (L57). Previous studies about immunostimulation activity by intake of insects were already mentioned in L57-61 (reference No. 12-16). 

Reviewer’s comment:

Lines 61-62: “but likely has functions for health management in cultured fish.”-> What does mean health management?

Our response:

We rewrote the sentence, as “Our study shows that MW is not only a protein source for aquaculture feed, but likely has functions for health optimization in cultured fish.”

Reviewer’s comment:

Material and methods:

It will be nice to start with

2.1. Feed ingredients

2.2. Experimental diets

2.3. Feeding trials

2.3.1 Growth performances

2.3.2 Challenge test with ..

2.4. Amino acid and fatty acid analysis

Of course, the part about the statistics is missing in the material and methods.

Our response:

    We rewrote this section according to your instruction as follows.

2.1. Feed Ingredients

2.2. Experimental diets

2.3. Feeding trials

2.3.1. Growth test

2.3.2. Challenge test with Edwardsiella tarda

2.4. Proximate composition, amino acid and fatty acid analysis

2.5. Statistical analysis

Reviewer’s comment:

Lines 71-73: “This defatting process was repeated several times until the crude lipids reached a comparable or lower level than fish meal and the defatted MW (MW meal) was obtained.”-> How did you compared the obtained defatted MW with the fish meal? 

Our response:

    We rewrote the sentence to clarify it, as “This defatting process was repeated several times until the crude lipids reached below 8 % and the defatted MW (DMW-1) was obtained.”

Reviewer’s comment:

Did you analyzed the feed ingredients, fish meal used and the most important the defatted mealworm?

Our response:

    We added the results of analysis of crude protein, crude fat, ash and amino acid composition in defatted mealworm and fish meal at Table 3 in the Results section.

Reviewer’s comment:

What was the subtract used for the growth of MW?

Our response:

   We added the subtract “wheat bran and vegetable wastes”.

Reviewer’s comment:

Did you used another defatted MW for the challenge test, if yes could you give another name?

Our response:

   We named defatted mealworm meal for the growth test “DMW-1”, and it for the challenge test “DMW-2”.

Reviewer’s comment:

Lines 81-86: could you develop more about the amino acid and fatty acid analysis, the chemicals, the internal standard….

Our response:

    We precisely rewrote the method.

Reviewer’s comment:

Line 88: could you stat and clarify that 2 experimental diets were produced, one for the growth of red seabreams (Table 1) and one for the challenge test (Table 2).

Our response:

    We added the statement at the front of “2.3.2 Challenge test with Edwardsiella tarda”. 

Reviewer’s comment:

Line 87: In the part of diet composition, a lot of data are missing, like the inclusion level and presentation of the diets; 65% FM is the control group and after that the fish meal was partially or completely replaced by defatted mealworms at ….

Our response:

   We added the information about diet compositions from Line 151-155 which you recommend to move to the Materials and Methods section.

Reviewer’s comment:

The title of the tables are not clear and there are not a footnotes? Example Table 1; Title: formulation, proximate composition of the diets fed to..

Our response:

    We changed the title and added the footnotes.

Reviewer’s comment:

Is it % on wet weight, dry ?

Our response:

   We mentioned it in the footnotes.

Reviewer’s comment:

footnote: explain what is the 65% FM, 25% MW….. and DHA is used as abbreviation but not defined before in the text.

Our response:

   We added the footnotes to explain the abbreviations, and the control and test diets.

Reviewer’s comment:

Why the non-essential amino acid taurine was added?

Our response:

   Lack of taurine in diet resulted in green liver syndrome and growth retardation. We added the explanation in L 188-189.

Reviewer’s comment:

How about the methionine and lysine, as they are deficient in IM, did you add those essential amino acids in the experimental diets?

Our response:

   Methionine and lysine in DMW-1 and DMW-2 are sufficient with dietary amino acid requirement estimates of red seabream (Foster and Ogata, 1998). We showed the essential amino acids in the diets in Table 1, and mentioned it in L 186-187.

Reviewer’s comment:

Line 100, could you remove artificially seed

Our response:

   We removed it.

Reviewer’s comment:

the growth tests are missing a lot of information, example during which time the trial was done, how long, the sampling how many, for each experience write again n=?

Our response:

   We already showed n=32, and measurement of FL and BW (all fish, 3 times, and so on).

Reviewer’s comment:

Again there are no information about the statistics analysis?

Our response:

    We added the section “2.5. Statistical analysis”. in L176-180. “2.5. Statistical analysis - Statistically significant differences between the control and test groups were identified by Steel-Dwass multiple comparison tests as a post hoc test after a Kruskal-Wallis test for growth performance, and a log-rank test with a Bonferroni correction for survival rate in the challenge test. Both tests were conducted with “R” software (https://www.r-project.org).”

Reviewer’s comment:

Results:

as you insect meal used in this study is defatted, maybe you could give a name like DMW.

Our response:

    We named defatted mealworm meal “DMW”.

Reviewer’s comment:

Line 135: you analyzed the amino acid and the fatty acid composition of the defatted mealworms used as feed ingredients for the growth trial or for the challenge test? 

Our response:

   We analyzed amino acids profile in both DMW-1 (growth trial) and DMW-2 (for challenge test), but fatty acids profile in the only mealworm for DMW-1, because we obtained DMW-2 in which fat content is too low. We showed the amino acid profiles in DMW-1, -2, and fish meal on Table 3.

Reviewer’s comment:

You analyzed the AA and FA but not the protein and the fat content?

Our response:

   We analyzed the content of crude protein, crude fat and ash. We showed them in Table 3.

Reviewer’s comment:

why did you use the reference number 6 ?

Our response:

   We deleted it.

Reviewer’s comment:

the fish meal in Table 3 is the same used to formulate the diets ?

Our response:

   Yes.

Reviewer’s comment:

The same with Table 4, the fatty composition of MW is the one used in the growth trial ? in addition the data of fish oil is it the one used as well in the experimental trial? how about the mealworm oil, did you analyze it also?

Our response:

   The fatty acid composition mealworm larvae are shown in Table 4. Mealworm oil is the oil fraction from the larvae in the defatting process. Thus, the fatty acid composition of mealworm oil is the same as that of mealworm larvae. We mentioned it in the footnotes. Fish oil on Table 4 was used for the experiment.

Reviewer’s comment:

Same comment for Table 3 and 4, the titles are not clear and the footnotes are missing.

Our response:

   We corrected the titles and added the footnotes.

Reviewer’s comment:

Line 149: Growth performances of red seabream fed diets with increasing replacement of FM with DMW.

Our response:

We corrected the titles.

Reviewer’s comment:

Line 151-155: “We tested 65% FM in comparison to 25% MW, 40% MW and 65% MW diets (containing 38%, 62% and 100% replacement of FM, respectively). Additionally, to assess the effect of the oil fraction from MW (MO), 65% MW + MO diets were tested. DHA was supplemented in the 65% MW + MO to be equivalent in n-3 HUFAs content with the 65% MW diet.” -> it should be in the material and methods part.

Our response:

   We moved the sentence to the materials and methods part (L 85-91).

Reviewer’s comment:

Line 155-157: “Diets containing MW showed a statistically significant tendency to promote FL and BW gain, depending on the increase in the MW content, with the 65% MW group showing the maximum growth (Figure 1; Table 156 5).” this sentence is unclear, if you could rephrase it, example a partial or complete replacement of FM with DMW in the diets of red seabream led to significantly increase the FL and BW compared to fish fed with the 65% FM diets. In addition for the BW, could explain also the differences observed between fish fed the % of DMW.

Our response:

   We change the sentence followed your instruction, as “A partial or complete replacement of FM with DMW-1 in the diets of red seabream led to significantly increase the FL compared to fish fed with the 65% FM diets (Figure 1; Table 5). In an addition for the FL, BW gain in fish increased as more DMW-1 was contained in the diets (Figure 1; Table 5).”

Reviewer’s comment:

The title for the Figure 1 need to changed, example Growth performances of

red seabream fed diets with increasing replacement of FM with DMW….

Our response:

      We changed the title as “Growth performance of red seabream fed diets with a partial or complete replacement of FM with DMW-1.”

Reviewer’s comment:

The same comment as before for the title of Table 5.

Table 5 and 6 could be in the same table. In Table 5 there are almost the same data as in Figure 1, I feel that there are repetition and no need to have the FL and BW three different times (with different calculation).

Our response:

       We change the title as “Fork Length and Body Weight gain, Feed intake, FCR and survival rate of red seabream fed diets with a partial or complete replacement of FM with DMW-1.” We deleted data about FL and BW. 

Reviewer’s comment:

Line 178-180: if you could rephrase the sentence.

Our response:

       We change the sentence as “Fish were fed with the control diet or diets containing 5% or 10 % DMW-2 for 56 days before the challenge. These diets did not lead to a significant effect on fish growth. After challenged with E. tarda, fish fed the diet including DMW-2 show higher survival ratio than control, and diets containing 10% DMW-2 resulted in highest survival than the others(Figure 2).” (L231-234).

Reviewer’s comment:

Figure2: could you put the legend for the figure (x mark, filled diamond and Open Square) on the side of the figure.

Our response:

   We put the figure legend on the side of the figure.

Reviewer’s comment:

Discussion:

My main concern is the discussion, as there is almost no discussion and this part need to be completed.

Our response:

   We rewrote the discussion, and believe that it is completed.

Reviewer’s comment:

Line 190-191: “High rates of insects in aquaculture feed have resulted in growth reduction in many cases, particularly in carnivorous fish species.” -> this is not true, as now many papers showed that is possible to include the high level of insect ingredients in the diets of carnivorous fish, such as salmonids, salmon and trout.

Our response:

   We checked the previous studies again, but no complete replacement of fish meal with mealworm larvae in fish feed for marine fish was reported. We added the detail of the studies (Piccolo, 2017; Gasco, 2016) (L253-260)

Reviewer’s comment:

Line 194-195: could you provide references.

Our response:

   We provided the citation, Henry et al. “Review on the use of insects in the diet of farmed fish: Past and future.” Anim. Feed Sci. Technol., 2015. They said that defatting insect meal might be advisable in a section “4.5.3. Defattening”

Reviewer’s comment:

Line 201: might work negatively?????

Our response:

   We change the sentence as “the oil fraction from MW was thought to give a negative effect for fish feed intake.” (L275)

Reviewer’s comment:

Line 202: which negative effect from intact MW larvae?

Our response:

   We change the sentence as “removes the negative effect for fish intake from intact MW larvae.” (L277-278)

Reviewer’s comment:

The references need to be checked, italic for the Latin name

Our response:

   We carefully checked the reference list and corrected it.